# Validity Evaluation of an Inertial Measurement Unit (IMU) in Gait Analysis Using Statistical Parametric Mapping (SPM)

**DOI:** 10.3390/s21113667

**Published:** 2021-05-25

**Authors:** Sangheon Park, Sukhoon Yoon

**Affiliations:** Motion Innovation Center, Korea National Sport University, Seoul 05541, Korea; ptl2503@knsu.ac.kr

**Keywords:** inertial measurement units, motion capture system, gait analysis, validity

## Abstract

Inertial measurement units (IMUs) are possible alternatives to motion-capture systems (Mocap) for gait analysis. However, IMU-based system performance must be validated before widespread clinical use. Therefore, this study evaluated the validity of IMUs using statistical parametric mapping (SPM) for gait analysis. Ten healthy males (age, 30.10 ± 3.28 years; height, 175.90 ± 5.17 cm; weight: 82.80 ± 17.15 kg) participated in this study; they were asked to walk normally on a treadmill. Data were collected during walking at the self-selected speeds (preferred speed, 1.34 ± 0.10 m/s) using both Mocap and an IMU. Calibration was performed directly before each gait measurement to minimize the IMU drift error over time. The lower-extremity joint angles of the hip, knee, and ankle were calculated and compared with IMUs and Mocap; the hip-joint angle did not differ significantly between IMUs and Mocap. There were significant differences in the discrete (max, min, and range of motion) and continuous variables (waveform: 0–100%) of the knee and ankle joints between IMUs and Mocap, particularly on the swing phase (*p* < 0.05). Our results suggest that IMU-based data can be used confidently during the stance phase but needs evaluation regarding the swing phase in gait analysis.

## 1. Introduction

Gait analysis is a cornerstone of movement-disorder evaluation. Human eyes can often overlook key information during clinical gait evaluation [1]. To avoid this, marker-based motion-capture (Mocap) systems can be used to provide objective assessments [2]. Mocap is used to identify variations in human movements, including walking, running, and cutting; it is the current gold standard for gait analysis [3,4,5,6,7]. However, its use is limited because of the following disadvantages: high cost, a lack of portability, need for expert operators, time-consuming setup and post-processing, and the need for patients to attend a specific laboratory [8]. Therefore, Mocap use is not widely used in clinical evaluation, and clinicians normally do not attempt to use objective kinematic information to assess human movement [9]. In the clinic, a thorough approach for quantifying joint kinematics is needed.

In the past couple of decades, quantitative gait analysis using portable Mocap has been made possible by technological developments in the field of Mocap [10]. Inertial measurement units (IMUs) are being used increasingly as possible solutions for collecting objective data for gait analysis in clinical settings [11]. IMU-based systems have the following advantages: low-cost, comport, portable, user-friendly, and suitable for outdoor use; moreover, they have a simple calibration process compared with Mocap [12,13]. Such IMU-based systems typically contain accelerometers, gyroscopes, and magnetometers. Accelerometers present a vertical axis that is in a static-orientation frame relative to gravity [14]. Gyroscopes track the sensor-orientation frame by integration over time (sensor XYZ axis) [15], and magnetometers detect the magnetic field vector relative to the Earth’s north-pointing magnetic field [16]. Using sensor-fusion algorithms, such as the complementary and Kalman filters, IMUs can track orientation with respect to a global coordination system [17,18]. Fusion algorithms provide the flexibility to select optimal state estimates (e.g., accelerometers combined with magnetometers fittingly estimate data during static movements, whereas gyroscopes present better data during dynamic movements [14]). However, concerns remain over various errors that occur in IMU-based systems. For example, IMU-based estimation suffers a drift error over time due to the integration of a gyroscope signal [19]. In addition, IMU-based data may be disturbed if nearby magnetic materials distort the local magnetic field [20]. Moreover, IMU-based kinematic position data are indirectly measured, unlike with Mocap [10]. Hence, IMU-based systems’ performance needs to be validated before they can become widespread in clinical gait analysis.

Several researchers have investigated the validity of IMU-based systems compared with Mocap [9,20,21,22,23,24,25]. Most of these studies indicate that IMU-based systems are useful for determining spatiotemporal variables and calculating ranges of motion (ROM). According to the previous studies, there was excellent agreement between the IMU-based and Mocap systems. Teufl and Miezal [20] found that the root mean squared error (RMSE) and range-of-motion error (ROME) of the joint angle in the sagittal plane were less than 1° between the IMU-based system and Mocap. Brice and Phillips [24] also mentioned a high level of agreement in the pelvis-related angle in the sagittal plane between the IMU-based system and Mocap. In addition, a previous study described the movements in closed chains as providing accurate results in IMU-based data [26]. However, three-dimensional (3D) kinematic measurements by IMUs have yet to show sufficient quality. In particular, studies have focused on comparing discrete variables calculated by IMU-based systems, including gait parameters (velocity, stride length, cadence, and others), peak value, and kinematic data at an event [9,22,23]. Differences in discrete variables may be overestimated or underestimated. Above all, before focusing on movement at a specific time point, it is critical to assess the entire movement in the clinic [27]; hence, IMU-based systems’ validity should be evaluated over the entire gait cycle.

Statistical parametric mapping (SPM) can be used as a method for analyzing the movement trajectory over the entire gait cycle [28]. SPM was originally used for neuroimaging, particularly functional magnetic resonance imaging (fMRI) [29]; however, it has recently been used to analyze kinematic data [28,30]. SPM is useful for examining the entire trajectory and is available for analyzing continuous data that change with time, in contrast with the scalar observations of discrete variables [27]. Studies have recently applied this method to the statistical analyses of kinematic and kinetic time series data [28,31,32]. Because it reflects the entire trajectory rather than just the movement at a specific time, it can offer a solution to the limitations of previous studies, which focused only on discrete variables. SPM offers a couple of significant advantages for biomechanists and movement scientists [33].

The present study aimed to evaluate the validity of an IMU using SPM for gait analysis. We aimed to test two hypotheses: (1) discrete variables do not show a significant difference between the IMU-based system and Mocap, similar to previous studies, and unlike previous studies, (2) continuous variables show a significant difference between the IMU-based system and Mocap.

## 2. Materials and Methods

### 2.1. Participants

The participants were 10 males with no history of lower-limb injury or pain and who did not experience even a slight injury within the past 6 months (age, 30.10 ± 3.28 years; height, 175.90 ± 5.17 cm; weight, 82.80 ± 17.15 kg). The Korea National Sports University Institutional Review Board approved this study (IRB No. KNSU 202104-047), and all participants provided written informed consent. According to Berner et al. (2020), a sample size of 9 to 15 is appropriate for this type of study [10].

### 2.2. Procedure and Data Collection

After a full warm-up and treadmill-adaptation period over 10 min [34], participants were asked to walk “as normally as possible” on an instrumented treadmill (Bertec, Columbus, OH, USA; 1000 Hz). Walking was performed at each user’s preferred speed (preferred speed, 1.34 ± 0.10 m/s). Data from a total of 10 strides were collected and used in the analysis.

For gait analysis using Mocap, all trials were recorded with eight infrared cameras (Oqus 300+, Qualiysis, Gothenburg, Sweden; 200 Hz) operated by Mocap software (Qualiysis track manager, Qualiysis, Gothenburg, Sweden). Cameras were positioned around the treadmill and calibrated using nonlinear transformation (NLT). The overall camera-reconstruction error was 0.15 mm for the camera calibration area. For gait analysis using the IMU-based system, we used four IMUs (myoMOTION Research Pro, Noraxon, Scottsdale, AZ, USA; 200 Hz) in this study. An IMU-based systems algorithm was provided by commercial software (MyoRESEARCH 3.10.64 [MR3]).

A total of 20 retro-reflective markers (9.5 mm diameter), 2 clusters, and 4 IMUs were simultaneously placed on participants’ bodies. Markers were placed on the right and left iliac crest, right and left anterior superior iliac spine, right and left posterior superior iliac spine, sacrum, right and left greater trochanter, thigh (segment marker), medial and lateral epicondyle, shank (segment marker), medial and lateral malleolus, toe, meta 1, meta 5, navicular bone, and heel (Figure 1). In addition, IMUs were attached on the pelvis (body area of the sacrum), elasticated straps on the thigh (frontal and distal half, where there is a lower amount of muscle displacement during gait), and shank (front and slightly medial to be placed along the tibia), and a foot (upper foot, slightly below the ankle) as a bandage (Figure 2).

We recorded a static Mocap trial to provide a baseline for the dynamic trials. At the same time, IMUs were calibrated on an anatomical standing pose for approximately 15 s; the calibration was performed just prior to gait measurement to minimize the drift error of IMUs over time. One experienced researcher attached all markers and IMUs to avoid the inter-rater measurement error, and medial markers were removed before collecting the data of the dynamic trials.

### 2.3. Data Processing and Analysis

The marker coordinates of Mocap were identified through automatic marker tracking using Qualiysis Track Manager (QTM). These coordinates were smoothened by filtering to reduce random noise. A zero-phase lag fourth-order low-pass Butterworth filter was applied with a cutoff frequency of 6 Hz [35]. Joint centers were calculated based on the positional data of attached markers. The medial/lateral malleolus and epicondyles were used to compute the midpoints as the ankle and knee-joint centers, respectively, and the Tylkowski method (1982) was used for the hip-joint center [36]. The lower rigid-body model defined using the proximal and distal joint centers was used in this study. The Cardan–Euler XY′Z′′ rotation sequence was performed to describe the relative orientation angles of the segments. Each lower-extremity joint angle was set to a positive value for flexion (dorsiflexion) and a negative value for extension (plantar flexion) with respect to the x-axis. The lower-extremity joint angle was analyzed in Visual 3D software (C-Motion, Germantown, MD, USA). Initial heel-strike and toe-off events were detected based on the ground-reaction force data with a threshold value set to 20 N by the instrumented treadmill.

The IMU-based body model for calculating joint angle was defined using MyoResearch 3 (MR3; Noraxon, Scottsdale, AZ, USA). The kinematics data were derived from relationships between different right-handed Cartesian coordinate systems (x-axis: pointing towards the top of the IMU along its length, y-axis: pointing to the left of the IMU, z-axis: pointing outwards perpendicular to IMU surface). The joint angle decomposition sequences in this software follow the recommendations of the International Society of Biomechanics (ISB) [37]. Moreover, the software performs a Kalman filter (robust-fusion algorithm) that is optimized for IMU-based data [10]. A more detailed description of the method can be found in the technology overview produced by Noraxon [38].

The lower-extremity joint angles analyzed in Visual 3D and MR3 were, respectively, exported and imported into MATLAB (R2016a, MathWorks, Natick, MA, USA). The maximum and minimum angles and ROM were calculated for the comparison of discrete variables during the gait cycle. To compare continuous variables (waveforms), we normalized each lower-extremity joint angle to the percentage of the gait cycle (101 data points, 0–100%). IMU-based and Mocap data were synchronized to the trigger signal sent by the Myosync system (Noraxon, Scottsdale, AZ, USA). MR3 software started the trigger signal automatically. IMU-based data were time-normalized and synchronized with Mocap data to enable a comparison of time series data.

### 2.4. Statistical Analysis

Statistical analysis was performed using SPSS (Version 21.0, SPSS Inc., Chicago, IL, USA) and MATLAB. A paired t-test was performed to verify differences in the discrete lower-extremity joint angle variables between the IMU system and Mocap. Moreover, a one-dimensional SPM (1D-SPM) paired *t*-test was performed to determine differences in these variables between IMUs and Mocap using open-source code (www.spm1d.org, accessed on 7 April 2021) in MATLAB [39]. The significance level was set at α = 0.05.

## 3. Results

### 3.1. Discrete Variables of the Lower-Extremity Joints during Walking

The difference in the hip-joint angles observed using the IMU-based and Mocap systems during walking was not significant (Table 1). However, the maximum knee-joint angle and ROM differed significantly between IMUs and Mocap in the sagittal plane (Table 1; *p* < 0.05). Moreover, the maximum and minimum ankle-joint angles were significantly different between IMUs and Mocap in the sagittal plane (Table 1; *p* < 0.05).

### 3.2. Continuous Variables of the Lower-Extremity Joints during Walking

The difference in the hip-joint angles recorded by the IMU-based and Mocap systems during walking was not significant (Figure 3; *p* > 0.05). However, the knee-joint angle differed significantly between IMU and Mocap measurements in the range from 70% to 90% of the phase (Figure 4; *p* < 0.05). Moreover, the ankle-joint angle differed significantly between IMUs and Mocap measurements in the ranges from 48% to 65% and from 79% to 100% of the phase (Figure 5; *p* < 0.05).

## 4. Discussion

The present study aimed to evaluate the validity of lower-extremity joint angle measurements using a commercial IMU-based system in comparison with Mocap. Mocap data were used as the gold standard for gait measurements. Discrete (max, min, and ROM) and continuous variables (waveform: 0–100%) during walking were quantified to demonstrate differences between the IMU-based system and Mocap.

No significant differences were found between the discrete variables of the datasets for the hip-joint angle during walking (Table 1, *p* > 0.05), which suggests that IMU-based outcomes are sufficiently sensitive to estimate the hip-joint angle during walking. In particular, there was an excellent agreement between the IMU-based and Mocap systems; the difference in hip-joint ROM was <1° in this study. This result was similar to that of previous studies comparing the IMU-based system against Mocap [9,20,21]. Teufl and Miezal [20] showed that the root mean squared error and range-of-motion error of the hip-joint angle in the sagittal plane was <1° between these two systems. Moreover, Brice and Phillips [24] found a high level of agreement in the pelvis-related angle in the sagittal plane between the IMU-based system and Mocap.

There were significant differences in the discrete variables of the knee and ankle joints (Table 1, *p* < 0.05). The maximum knee-flexion angle and ROM measured by IMUs were greater than those measured by Mocap; this angle typically occurs on the swing phase during walking. The overestimation of this angle propagates into an overestimation of ROM. This suggests that the IMU-based measurement of maximum knee-flexion angle as a discrete variable must be carefully considered on the swing phase during walking. Moreover, the maximum ankle dorsi and plantar flexion angles measured by IMUs tended to be underestimated overall in the ankle plantarflexion direction. In general, the maximum ankle dorsiflexion angle occurs between the terminal-stance (31–50%) and pre-swing phases (50–62%), whereas the maximum ankle plantarflexion angle occurs between the pre-swing (50–62%) and initial-swing (62–75%) phases [40]. In the cases of knee and ankle joints, caution should be taken when measuring the joint angle with the IMU-based system after the terminal-stance phase (31–50%), particularly during the swing phase (50–100%).

An important difference was observed in the lower-extremity joint angle using 1D-SPM between the IMU-based system and Mocap. There was consistently no significant difference between the IMU-based and Mocap measurements of continuous hip-joint variables during walking (Figure 3, *p* > 0.05), suggesting that the IMU-based system had a high degree of validity for measuring hip joints. However, in this study, the measured knee-joint angles differed significantly during the swing phase (70–90%; Figure 4, *p* < 0.05). A previous study described movements in closed chains as providing more accurate results [26]. In the present study, the knee-joint angle measured by IMU was valid during the stance phase but not during the swing phase. This is because gravitational acceleration is used to calculate joint angles during the stance phase since stationary IMUs measure the gravitational acceleration vector directly. It thus allowing obtaining an absolute measurement of the inclination angles of the segment to which the IMU was attached during stance phase. This consistent result suggests that the IMU-based system needs refinement to calculate the lower-extremity joint angle in open chains.

Moreover, the ankle-joint angle was significantly different between the IMU-based system and Mocap during the pre-swing (48–65%) and terminal-swing phases (79–100%; Figure 5, *p* < 0.05) [40]. There were significant differences in the discrete variables of the ankle-joint angle. Here, the ankle-joint showed significant differences in the maximum dorsiflexion and plantarflexion angles between the IMU-based and Mocap systems. This finding is consistent with previous studies that revealed differences between the IMU-based system and Mocap [5,21,41]. The IMU-based system may exhibit decreased validity from the proximal to distal regions (hip, knee, and ankle), as demonstrated by Poitras and Dupuis [26]. Cooper and Sheret [42] found that the accuracy of IMU-based data may decrease with increasing movement speed; as mentioned above, the dorsiflexion and plantarflexion angles for evaluating ankle joints should be considered valid in the swing phase.

Studies have suggested that the measurement of the joint angle between the IMU-based system and Mocap system data presents a reasonably good agreement [7,12,17,21,22,25,42,43,44,45]. Moreover, the findings of the present study showed high validity of IMU measurements of the hip-joint angle and of the overall waveforms of the knee and ankle joint angles on the sagittal plane. However, although the IMU-based data tended to follow a similar waveform to the Mocap data, the IMU-based data exhibited a significant error during the specific phase. Therefore, additional research is needed to develop methods for the IMU-based system during high-speed and complex movements on a specific phase, particularly on the swing phase in gait analysis. We propose some recommendations for IMU users: the optimal position of IMU, maximum movement speed for correct data collection, and caution for movement complexity. Moreover, evaluating the validated sensor-fusion method in a setup should be considered. Furthermore, because gait analysis is a very important and essential index in the clinic, it is necessary to satisfy the accuracy of the joint angle during walking. Our findings may suggest a possible direction for a more valid analysis of IMU-based data in human-gait studies. Based on the present study, we recommend that lower-extremity joint angle data using IMUs on the swing phase should be cautiously considered during gait analysis. However, an IMU-based system can capture several valid gait outcomes for discrete and continuous variables.

A limitation of the present study was that only lower-extremity joints (rather than joints across the entire body) were examined. Further studies with IMUs are needed to determine the validity of this system for whole-body joint kinematic data. Additionally, a single commercialized IMU was analyzed in this study; thus, it is necessary to evaluate the validity of other diverse IMUs and present the advantages and disadvantages between various IMUs. Future studies should especially focus on different fusion algorithms, which may potentially yield different results.

## 5. Conclusions

Our findings indicated an excellent agreement of the lower-extremity joint angle (hip, knee, and ankle joint) measured by IMUs with those by Mocap, particularly regarding the hip-joint angle. However, errors in the IMUs’ knee and ankle-joint measurements increased during the swing phase. This error affects discrete variables, such as the max joint angle, min joint angle, and ROM. Therefore, we suggest that IMU-based data can be confidently used overall on the stance phase but needs a critical evaluation with respect to the swing phase (70–90% in the knee joint and 48–65% and 79–100% in the ankle joint) in gait analysis.

## Figures and Tables

**Figure 1 sensors-21-03667-f001:**
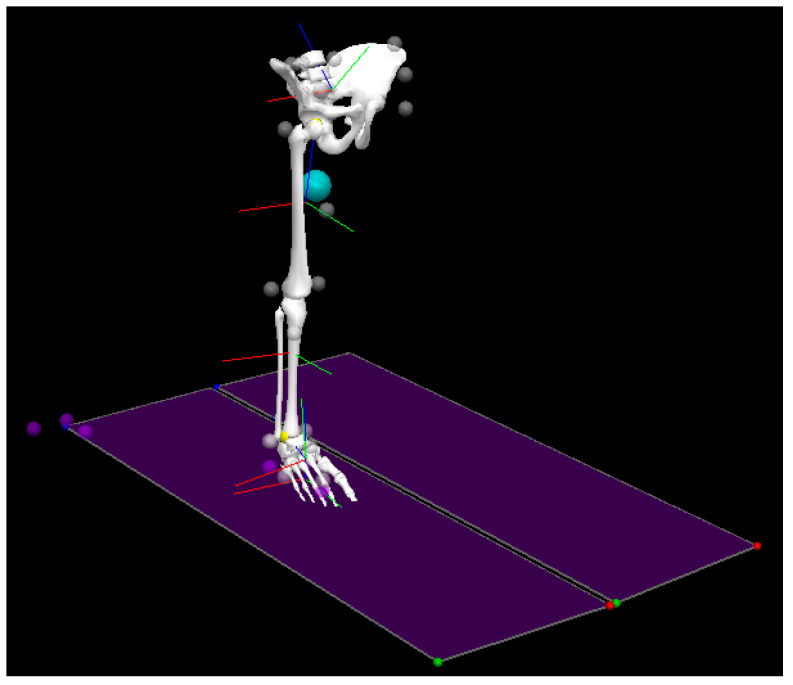
Attachment of markers on the lower extremity (the axis of angular displacement was defined as follows: red line: positive x-axis vector; green line: positive y-axis vector; blue line: positive z-axis vector; white circle: attachment of markers) by Visual 3D professional v6.03.4.

**Figure 2 sensors-21-03667-f002:**
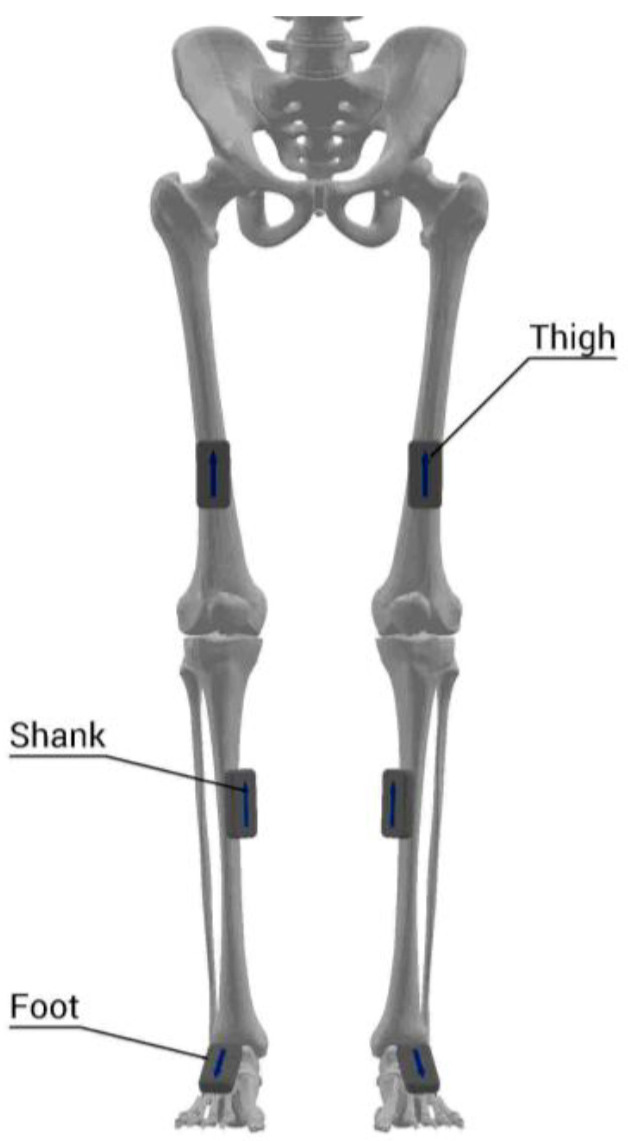
Attachment of IMUs on the lower extremity (the axis of angular displacement was defined to be the same as the axis of angular displacement of the Mocap-based coordination system) using MR3 3.16.

**Figure 3 sensors-21-03667-f003:**
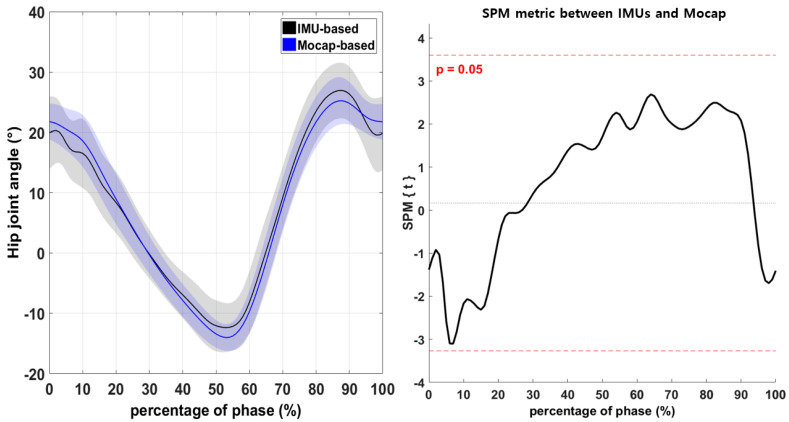
Comparison of the hip-joint angle between the IMU-based and Mocap systems according to one-dimensional SPM. Horizontal dashed lines illustrate significant differences at α = 0.05, and the result is visualized as a waveform.

**Figure 4 sensors-21-03667-f004:**
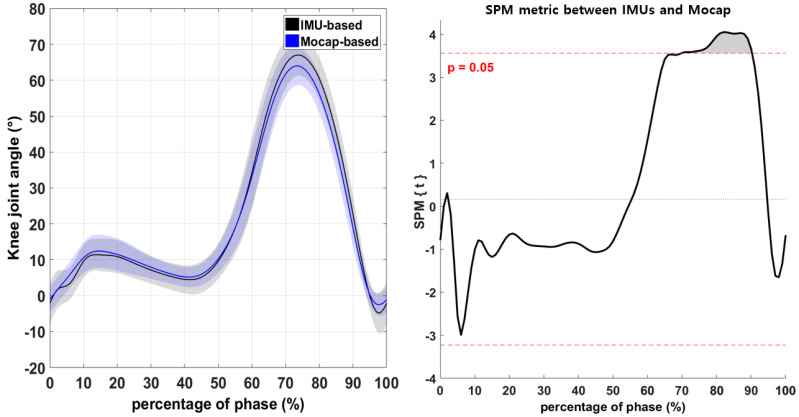
Comparison of the knee-joint angle between the IMU-based and Mocap systems according to one-dimensional SPM. Horizontal dashed lines illustrate significant differences at α = 0.05, and the result is visualized as a waveform (the gray area in the figure on the right indicates significant differences).

**Figure 5 sensors-21-03667-f005:**
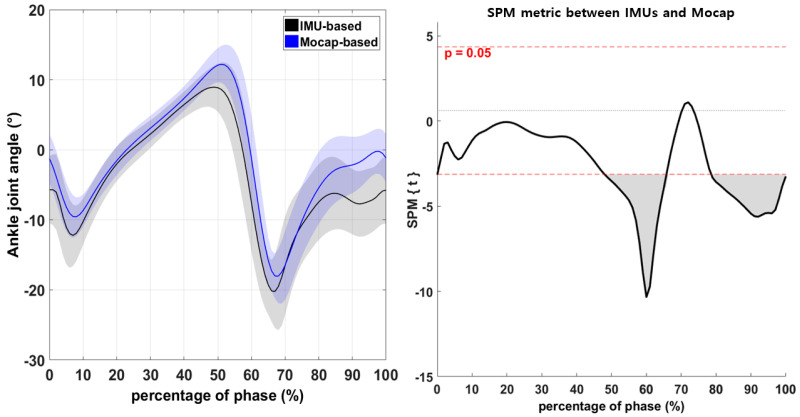
Comparison of the ankle-joint angle between the IMU-based and Mocap systems according to one-dimensional SPM. Horizontal dashed lines illustrate significant differences at α = 0.05, and the result is visualized as a waveform (the gray area in the figure on the right indicates significant differences).

**Table 1 sensors-21-03667-t001:** Comparison of lower-extremity joint angles between the IMU and Mocap systems.

(Unit: deg)	Variable	System	Mean ± SD	t (*p*)
Hip-joint angle (+ flexion / − extension)	Max	IMU	27.18 ± 4.66	1.88 (0.09)
		Mocap	25.70 ± 3.85	
	Min	IMU	−12.97 ± 3.83	1.83 (0.10)
		Mocap	−14.41 ± 2.23	
	ROM	IMU	39.89 ± 3.81	0.01 (1.00)
		Mocap	39.88 ± 3.22	
Knee-joint angle (+ flexion /− extension)	Max	IMU	67.66 ± 5.79	3.29 (0.01) *
		Mocap	64.58 ± 5.21	
	Min	IMU	−5.66 ± 5.79	−1.84 (0.10)
		Mocap	−3.18 ± 3.11	
	ROM	IMU	72.67 ± 5.34	7.07 (0.01) *
		Mocap	67.20 ± 4.66	
Ankle-joint angle (+ dorsi flexion /− plantar flexion)	Max	IMU	9.63 ± 2.90	−5.33 (0.01) *
		Mocap	12.66 ± 2.71	
	Min	IMU	−23.16 ± 5.09	−5.20 (0.01) *
		Mocap	−19.44 ± 3.79	
	ROM	IMU	31.84 ± 5.75	0.15 (0.89)
		Mocap	31.71 ± 4.97	

* indicates a significant difference; IMU: Inertial Measurement Unit; Mocap: Motion capture system; ROM: Range of Motion.

## Data Availability

Not applicable.

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
