# Peer review of "Validity Evaluation of an Inertial Measurement Unit (IMU) in Gait Analysis Using Statistical Parametric Mapping (SPM)"

_sensors, 2021, doi:10.3390/s21113667_

Round 1

Reviewer 1 Report

This is the study which evaluates the accuracy of joint angle pattern during gait motion measured by IMU. The difference between joint patterns calculated using motion capture and IMU is evaluated through gait cycle in addition to representative parameters such as peak joint angles. The evaluation of joint angle estimation using IMU is the hot issue in the field of gait analysis and this study seems helpful to validate the gait analysis using IMU. However, there are several concerns about the organization and logic of this article.

In this study, it seems that the t-test is used not only to judge the difference between joint patterns but also to claim the agreement between them (i.e. the beginning of the fourth paragraph of 4. discussion). However, the large p-value only means that the difference is not sufficient to reject null hypothesis. The authors should use ICC or other statistical method to claim the agreement between mocap and IMU. I think it is fair that only the difference of knee and ankle angle during swing phase is mentioned.

In this study, as many related studies, the performance of “IMU” is discussed. However, the comparison is performed using only single IMU product. Thus, it is difficult to generalize the conclusion of this study to the other IMU products. It is more useful to determine the evaluation method of the accuracy of IMU gait sensors which can be generally used.

It seems obvious that the acceleration vector of leg swing is added to the gravity vector and interferes the joint angle estimation using IMU. Thus, it is anticipated that the accuracy of IMU joint angle estimation decreases when used for overground walking. This concern should be added to the limitation.

It seems Fig. 3 is not necessary. It’s too basic.

Author Response

Dear Editor,

We thank you very much for your time and consideration on our manuscript titled “Validity evaluation of an inertial measurement unit (IMU) in gait analysis using statistical parametric mapping (SPM)”

Below we address Reviewer’s comments and our response to the comments as well as the changes that we made to our manuscript according to Reviewer’s reports. The original Reviewer’s comments are provided in black color, whereas our responses are given in blue.

We believe that these modifications have strengthened the manuscript, and hope that our revised manuscript is acceptable for publication in Sensors.

Thank you again for your attention and consideration.

Sincerely,

Authors

 Comments and Suggestions for Authors

This is the study which evaluates the accuracy of joint angle pattern during gait motion measured by IMU. The difference between joint patterns calculated using motion capture and IMU is evaluated through gait cycle in addition to representative parameters such as peak joint angles. The evaluation of joint angle estimation using IMU is the hot issue in the field of gait analysis and this study seems helpful to validate the gait analysis using IMU. However, there are several concerns about the organization and logic of this article.

Review: In this study, it seems that the t-test is used not only to judge the difference between joint patterns but also to claim the agreement between them (i.e. the beginning of the fourth paragraph of 4. discussion). However, the large p-value only means that the difference is not sufficient to reject null hypothesis. The authors should use ICC or other statistical method to claim the agreement between mocap and IMU. I think it is fair that only the difference of knee and ankle angle during swing phase is mentioned

: Thank you for your valuable comments. You are absolutely right. It is very critical comment, I agree. Because this study aimed to provide an evidence if the clinician can use IMU sensors directly in clinical field or not, we investigated the differences between Mocap and IMUs with a simple statistical method, such as t-test as a first step. The reliability using ICC or other statistical method to claim the agreement between two systems will be evaluated in the following studies in the near future. Please, looking forward to the paper (connected to the present article) that we will submit continuously. Thank you.

Review: In this study, as many related studies, the performance of “IMU” is discussed. However, the comparison is performed using only single IMU product. Thus, it is difficult to generalize the conclusion of this study to the other IMU products. It is more useful to determine the evaluation method of the accuracy of IMU gait sensors which can be generally used.

: Thank you for your thoughtful comment. The produce we evaluated in this study was Noraxon, and it is one of the generally used IMU products in the clinical including Xsens and IMeasureU. We believe that evaluation of these products will contribute to producing diverse and meaningful results and eventually will become a solution (and more approachable method) for collecting objective information for gait analysis in clinical settings. Page 10, Line 283-293. According to the reviewer’s opinions, this part was added as a limitation.

Review: It seems obvious that the acceleration vector of leg swing is added to the gravity vector and interferes the joint angle estimation using IMU. Thus, it is anticipated that the accuracy of IMU joint angle estimation decreases when used for overground walking. This concern should be added to the limitation.

: Thank you for your suggestions. Page 10, Line 283-293. According to the reviewer’s opinions, this part was revised.

Review: It seems Fig. 3 is not necessary. It’s too basic.

: Appreciate your suggestion. According to the reviewer’s opinions, this part was deleted.

Reviewer 2 Report

The manuscript is devoted to the research concerning the application of inertial measurement units for gait analysis. The topic is interesting and actual.

The manuscript is well structured; it contains all sections for this type of publication. The abstract briefly reflects the content of the manuscript. However, to my mind, the manuscript contains some shortcomings which should be corrected. 

  1. The English grammar should be corrected. The manuscript contains some mistakes and incorrectness.
  2. The section Introduction contains the analysis of related works too. To my mind, at the end of this section, it will be better to allocate the unsolved part of the general problem. Then, it is necessary to add the Problem Statement.
  3. The conclusion section should be extended to. This section should reflect briefly the obtained results and the perspectives of their further application.

Author Response

Dear Editor,

We thank you very much for your time and consideration on our manuscript titled “Validity evaluation of an inertial measurement unit (IMU) in gait analysis using statistical parametric mapping (SPM)”

Below we address Reviewer’s comments and our response to the comments as well as the changes that we made to our manuscript according to Reviewer’s reports. The original Reviewer’s comments are provided in black color, whereas our responses are given in blue.

We believe that these modifications have strengthened the manuscript, and hope that our revised manuscript is acceptable for publication in Sensors.

Thank you again for your attention and consideration.

Sincerely,

Authors

The manuscript is devoted to the research concerning the application of inertial measurement units for gait analysis. The topic is interesting and actual.

The manuscript is well structured; it contains all sections for this type of publication. The abstract briefly reflects the content of the manuscript. However, to my mind, the manuscript contains some shortcomings which should be corrected.

Review: The English grammar should be corrected. The manuscript contains some mistakes and incorrectness.

: Thank you for the quality comment. According to the reviewer's opinions, the English grammar and mistakes in this paper were double-checked overall (full article) by a professional English editing service. Thank you.

Review: The section Introduction contains the analysis of related works too. To my mind, at the end of this section, it will be better to allocate the unsolved part of the general problem. Then, it is necessary to add the Problem Statement.

: Thank you for your thoughtful comment. According to the reviewer's opinion, the part of introduction was modified additionally on Page 2, Line 57-73.

Review: The conclusion section should be extended to. This section should reflect briefly the obtained results and the perspectives of their further application.

: Appreciate your thorough review. According to the reviewer’s opinions, this part was revised on Page 10, Line 296-304.

Reviewer 3 Report

The paper presents the results of a validation study of an IMU-based motion capture system used for gait analysis. Flexion/extension angles for the hip and knee joints and dorsi/plantarflexion angles for the ankle joint are recorded using IMUs, which is then compared to joint angles obtained from an optoelectronic motion capture system as gold standard. Similar and validation studies have been presented previously. The main novelty that the authors propose in this study is the use of statistical parameter mapping (SPM) to analyse the validity of the continuous joint angle trajectories.

The presentation of the paper is lacking, in particular regarding the method and interpretation of the results. This makes it impossible to judge the validity of the proposed method, and must be addressed by the authors in a revision. Overall, the quality of the writing and the presentation must be improved. I will list my main concerns in the comments below and recommend that the authors address these in a major revision.

Major comments:

  1. The title is not specific enough. I think it should contain “gait analysis” or “joint angles” since this is the application that is studied.
  2. (line 99) How long is the warm-up and treadmill-adaption period?
  3. Why are only the hip/knee flexion/extension and ankle dorsi/plantarflexion angles studied? There is no mention about abduction/adduction and internal rotation angles. Why were these angles excluded? Not including these angles reduces the usefulness of the study.
  4. How are the axes of rotation of the joints defined in the IMU coordinate systems? This will greatly affect the accuracy of the joint angles. Depending on which approach that is used it might also be more or less sensitive to the sensor placement on each segment and the calibration pose/movement that is used. It would be interesting to see how much the joint angles change between different trials/subjects when the IMUs have to be removed and placed again.
  5. The use of SPM is the main novelty of the paper but it is not sufficiently described. The main results presented in Figures 4-6 are impossible to interpret without an adequate description of the SPM metric that is used. These figures are also mislabeled saying e.g. “Ankle joint angle by SPM” when the figure does not present an angle but rather the SPM metric which compares IMU to Mocap.
  6. In Table 1. It is hard to understand how some values can be significant when others are not. For example Knee-joint angle Max is significant and Min is not even though the differences in the mean values are about the same and difference in standard deviation is greater for Min than for Max. Are these values correct?

Minor comments:

  1. Figure 3 is not very informative. It would be sufficient to say that the IMU and Mocap systems are synchronized by a trigger signal in the software you used.
  2. Wrong citation styles or missing citations:
    1. line 87: Berner et al. (2020)
    2. line 129: Tylkowski’s method (1982)
    3. line 159: The URL for spm1d could use a citation.
  3. Line 264: Why is an outdoor experiment needed to generalize the findings?
  4. I strongly suggest that the authors ask someone to review the quality of the writing and English language. Here are some sentences/phrases that I recommend changing:
    1. line 20: “.. must be considered on the swing phase ..”.
    2. line 34: “stern approach” can have an awkward interpretation.
    3. line 51: “integrating a gyroscope” better to say that you integrate a gyroscope signal to be clear.
    4. line 61: “miss a significant or overstate a certain difference”.
    5. line 69: “scalar observations of discrete variables” I think “scalar” should be dropped here.
    6. line 74: “powerful strengths”.
    7. lines 108-109: The sentence describing how the IMUs are attached is hard to read.
    8. Figure 1 caption: Hard to read, missing spaces.
    9. lines 145-146: I do not think that “normalized” is correct here, you extracted joint angles for each gait cycle and computed averages/standard deviations.
    10. line 201: “accords” better to use “agrees”.
    11. lines 270-272: First sentence in the conclusions is very hard to read.

Author Response

Dear Editor,

We thank you very much for your time and consideration on our manuscript titled “Validity evaluation of an inertial measurement unit (IMU) in gait analysis using statistical parametric mapping (SPM)”

Below we address Reviewer’s comments and our response to the comments as well as the changes that we made to our manuscript according to Reviewer’s reports. The original Reviewer’s comments are provided in black color, whereas our responses are given in blue.

We believe that these modifications have strengthened the manuscript, and hope that our revised manuscript is acceptable for publication in Sensors.

Thank you again for your attention and consideration.

Sincerely,

Authors

The paper presents the results of a validation study of an IMU-based motion capture system used for gait analysis. Flexion/extension angles for the hip and knee joints and dorsi/plantarflexion angles for the ankle joint are recorded using IMUs, which is then compared to joint angles obtained from an optoelectronic motion capture system as gold standard. Similar and validation studies have been presented previously. The main novelty that the authors propose in this study is the use of statistical parameter mapping (SPM) to analyze the validity of the continuous joint angle trajectories.

Review: The presentation of the paper is lacking, in particular regarding the method and interpretation of the results. This makes it impossible to judge the validity of the proposed method, and must be addressed by the authors in a revision. Overall, the quality of the writing and the presentation must be improved. I will list my main concerns in the comments below and recommend that the authors address these in a major revision.

: Thank you for your thorough review. Additionally, according to the reviewer's opinions, the English grammar and mistakes in this paper were double-checked overall (full article) by a professional English editing service. Thank you.

Review: The title is not specific enough. I think it should contain “gait analysis” or “joint angles” since this is the application that is studied.

: Thank you for your valuable suggestion. According to the reviewer’s opinion, the word “gait analysis” was added to the Title.

Review: (Line 99) How long is the warm-up and treadmill-adaption period?

: Thank you for your comment. The full-warm-up and treadmill-adaptation period was set over 10 mins according to the previous study (Meyer, Killeen, Easthope, Curt, Bolliger, Linnebank, et al. 2019) and this was added on Page 3, Line 101

Review: Why are only the hip/knee flexion/extension and ankle dorsi/plantarflexion angles studied? There is no mention about abduction/adduction and internal rotation angles. Why were these angles excluded? Not including these angles reduces the usefulness of the study.

: You are absolutely right. It is a critical comment. Because this study was focused on the participants without any gait-related diseases, hip/knee flexion/extension and ankle dorsi/plantarflexion angles were analyzed which are commonly used in general population. The variables such as abduction/adduction and internal rotation angles are mainly used to detect the gait differences in the patients with certain diseases such as stroke, ACL rupture, or chronic ankle instability. It is because that the values of abduction/adduction and internal rotation angles become significant enough to depict when injured or paralyzed. (Please refer to the figure below; Hewett et al., 2005) Thus, these variables may be analyzed in the future studies focusing on the patients with certain diseases.

Hewett, T. E., Myer, G. D., Ford, K. R., Heidt Jr, R. S., Colosimo, A. J., McLean, S. G., ... & Succop, P. (2005). Biomechanical measures of neuromuscular control and valgus loading of the knee predict anterior cruciate ligament injury risk in female athletes: a prospective study. The American journal of sports medicine, 33(4), 492-501.

Review: How are the axes of rotation of the joints defined in the IMU coordinate systems? This will greatly affect the accuracy of the joint angles. Depending on which approach that is used it might also be more or less sensitive to the sensor placement on each segment and the calibration pose/movement that is used. It would be interesting to see how much the joint angles change between different trials/subjects when the IMUs have to be removed and placed again.

: Thank you for your thoughtful comment. It is a wonderful and useful idea. It is worth to consider the sensor placement and calibration pose/movement and will be useful to improve the usage of sensors in the field. There are two representative calibration methods such as standing and sitting and the IMU sensors company (in this article) is currently developing another calibration method for specific analysis. As you suggested, it would be valuable to see the different calibration methods.

The sensor placement locations on each segment followed the instructions according to the commercialized IMU company manual. The details were explained on Page 3, Line 118-122.

Moreover, the IMU coordination systems were defined on Page 4, Line 153-160. According to the reviewer’s opinions, this part was added and revised.

Review: The use of SPM is the main novelty of the paper but it is not sufficiently described. The main results presented in Figures 4-6 are impossible to interpret without an adequate description of the SPM metric that is used. These figures are also mislabeled saying e.g. “Ankle joint angle by SPM” when the figure does not present an angle but rather the SPM metric which compares IMU to Mocap.

: Thank you for your comment. According to the reviewer’s opinion, the titles and captions were fully revised in Figure 3-5.

Review: In Table 1. It is hard to understand how some values can be significant when others are not. For example Knee-joint angle Max is significant and Min is not even though the differences in the mean values are about the same and difference in standard deviation is greater for Min than for Max. Are these values correct?

: Thank you for your detailed comment. Yes, they are correct. The mean values of the Min Knee joint angle should be greater than the standard deviation. The reason is that the Min value of the knee joint angle represents the maximum extension, and the mean values should be close to 0 degree because the knee angle was set at 0 degree on standing calibration in this study.

Review: Figure 3 is not very informative. It would be sufficient to say that the IMU and Mocap systems are synchronized by a trigger signal in the software you used.

: Thank you for your comment. According to the reviewer’s opinions, this part was deleted.

Review:

line 87: Berner et al. (2020) / line 129: Tylkowski’s method (1982) / line 159: The URL for spm1d could use a citation.

: Appreciate your thorough review. According to the reviewer’s opinions, these parts were revised.

Review: Line 264: Why is an outdoor experiment needed to generalize the findings?

: Thank you for your comment. It is because the gravitational vector is added to the acceleration vector of the leg swing during the swing phase, which interferes with the joint angle estimation using the IMU. A decrease in the accuracy of IMU-based data estimation is anticipated when used for overground gait analysis. The reasons were added on Page 10, Line 283-293.

Review:

line 20: “.. must be considered on the swing phase .”. / line 34: “stern approach” can have an awkward interpretation. / line 51: “integrating a gyroscope” better to say that you integrate a gyroscope signal to be clear. / line 61: “miss a significant or overstate a certain difference”. / line 74: “powerful strengths”. / Figure 1 caption: Hard to read, missing spaces. / line 201: “accords” better to use “agrees”.

: According to the reviewer’s opinions, these parts were revised all.

Review: lines 108-109: The sentence describing how the IMUs are attached is hard to read.

: Page 4, 116-119. According to the reviewer’s opinions, this part was revised all.

Review: lines 270-272: First sentence in the conclusions is very hard to read.

: According to the reviewer’s opinions, conclusion was overall revised.

Review: I strongly suggest that the authors ask someone to review the quality of the writing and English language.

: Thank you for the quality comment. According to the reviewer's opinions, the English grammar and mistakes in this paper were double-checked overall (full article) by a professional English editing service. Thank you.

Round 2

Reviewer 3 Report

1) The written quality of the paper has improved after the revision. I am mostly satisfied but there are some typos still. For example "ferived" on line 154. I recommend that the authors again do a thorough spell check.

2) Figure 4 is wrong. It shows the Hip joint angle again. The figure for the knee joint was in the original draft.

3) The text added between lines 284-293 and on 299-300 contains some significant errors. Specifically, regarding the motivation for outdoor experiments, which say "It is because the gravitational vector is added to the acceleration vector of the leg swing during the swing phase, which interferes with the joint angle estimation using the IMU". First of all, this does not at all motivate that outdoor experiments should be performed. The gravitational acceleration vector is constant in any local environment. The motivation makes it sound like outdoors/indoors conditions would affect the gravitational acceleration, which is simply wrong. The correct statement would be to say that gravitational acceleration is used to calculate joint angles during quasi-stationary parts of the gait phase, since stationary IMUs then measure the gravitational acceleration vector directly. This can then be used to get an absolute measurement of the inclination angles of the segment to which the IMU is attached. However, again this does not motivate why outdoor experiments need to be performed. It would make sense to be outdoors if your method for computing joint angles rely on magnetometer measurements, since these are typically unreliable indoors. Magnetometer measurements should not have a big inpact on the joint angles that are studied in this paper since these joint angles are in the sagittal/coronal planes. Internal rotation angles, which are more aligned with the transversal planes would rely more on accurate heading measurements from e.g. a magnetometer.

I would like the authors to carefully reconsider their statement and provide a technically correct motivation.

4) The method description could be improved by adding more details about the sensor fusion algorithm used to calculate the joint angles. Currently, all that is said is that a Kalman filter is used for this (line 159) inside the software that comes with the IMUs. At least, I recommend that the authors contact the IMU manufacturer to ask if they have publications or patents for their method to calculate joint angles and add it as a reference. Different fusion algorithms can yield different results, so it is really the combination of the IMUs and the algorithm that is validated in this paper. I also recommend that the authors add a section to the discussion regarding the possibility of different fusion algorithms yielding different results.

Author Response

Dear Editor,

We thank you very much for your time and consideration on our manuscript titled “Validity evaluation of an inertial measurement unit (IMU) in gait analysis using statistical parametric mapping (SPM)”

Below we address Reviewer’s comments and our response to the comments as well as the changes that we made to our manuscript according to Reviewer’s reports. The original Reviewer’s comments are provided in black color, whereas our responses are given in blue.

We believe that these modifications have strengthened the manuscript, and hope that our revised manuscript is acceptable for publication in Sensors.

Thank you again for your attention and consideration.

Sincerely,

Authors

1) The written quality of the paper has improved after the revision. I am mostly satisfied but there are some typos still. For example "ferived" on line 154. I recommend that the authors again do a thorough spell check.

: Thank you for your review in detail. The typo “ferived” has been changed to “derived”.

2) Figure 4 is wrong. It shows the Hip joint angle again. The figure for the knee joint was in the original draft.

: Thank you for your thorough review and noticing the critical point. The part seems to have mistakenly presented the angle of hip as the angle of the knee joint during editing. It is purely my mistake and Figure 4 was fully revised.

3) The text added between lines 284-293 and on 299-300 contains some significant errors. Specifically, regarding the motivation for outdoor experiments, which say "It is because the gravitational vector is added to the acceleration vector of the leg swing during the swing phase, which interferes with the joint angle estimation using the IMU". First of all, this does not at all motivate that outdoor experiments should be performed. The gravitational acceleration vector is constant in any local environment. The motivation makes it sound like outdoors/indoors conditions would affect the gravitational acceleration, which is simply wrong. The correct statement would be to say that gravitational acceleration is used to calculate joint angles during quasi-stationary parts of the gait phase, since stationary IMUs then measure the gravitational acceleration vector directly. This can then be used to get an absolute measurement of the inclination angles of the segment to which the IMU is attached. However, again this does not motivate why outdoor experiments need to be performed. It would make sense to be outdoors if your method for computing joint angles rely on magnetometer measurements, since these are typically unreliable indoors. Magnetometer measurements should not have a big inpact on the joint angles that are studied in this paper since these joint angles are in the sagittal/coronal planes. Internal rotation angles, which are more aligned with the transversal planes would rely more on accurate heading measurements from e.g. a magnetometer.

I would like the authors to carefully reconsider their statement and provide a technically correct motivation.

: I absolutely agree with your opinion. As the reviewer said, the gravitational acceleration vector is literally the same in indoors and outdoors environments. Considering the concerns of the reviewer’s opinions, we have deleted the motivation for outdoor experiments. In addition, appreciate your opinion regarding the swing phase and gravitational acceleration. It was added in the discussion in line 251-256. Thank you very much.

4) The method description could be improved by adding more details about the sensor fusion algorithm used to calculate the joint angles. Currently, all that is said is that a Kalman filter is used for this (line 159) inside the software that comes with the IMUs. At least, I recommend that the authors contact the IMU manufacturer to ask if they have publications or patents for their method to calculate joint angles and add it as a reference. Different fusion algorithms can yield different results, so it is really the combination of the IMUs and the algorithm that is validated in this paper. I also recommend that the authors add a section to the discussion regarding the possibility of different fusion algorithms yielding different results.

: Fusion algorithms, as the reviewers said, are a very important part of this study. When conducting this research, I enquired the company about the algorithm but unfortunately, I did not receive an answer from them. Instead, we managed to find the information about the built-in algorithms from https://www.noraxon.com/noraxon-download/imu-technology-overview/ and it was added as a reference (line 159-161, ref no. [10], [38]). Moreover, the limitation about using a single machine (algorithms) was explained and the necessity of validation on the different fusion algorithms was suggested (line 289-293). Thank you for your great review.